# Effects of electroconvulsive therapy on hippocampal longitudinal axis and its association with cognitive side effects

## Abstract

**Background** Electroconvulsive therapy (ECT)-mediated hippocampal volumetric increase is consistently reported, though its clinical relevance remains debated. This study evaluates if ECT-related cognitive side effects are associated with regional volumetric changes along the hippocampal longitudinal axis.
**Methods** Longitudinal T1-weighted MRI scans in 435 patients (54.0 ± 15.0 years, 261 female) with major depression from the Global ECT-MRI Research Collaboration (GEMRIC) were used to measure changes in right global and longitudinal axis hippocampal subdivisions (head, body, tail) from baseline to post-treatment. Cognitive side effects were evaluated using pre-to-post treatment changes in two different verbal fluency tests available for 124 patients. Electric field modelling was applied to explore whether the regional hippocampal electric field strength related to individual changes in cognitive performance.
**Results** Global hippocampal enlargement is observed pre-to-post ECT ($p_{FDR} < 0.001$), but enlargement of the hippocampal head significantly exceeds the volumetric change in the hippocampal body and tail ($p_{FDR} < 0.001$). Volumetric expansion of the hippocampal body and tail significantly associates with reduced verbal fluency scores ($p_{FDR} < 0.05$). Moreover, volumetric reduction of the hippocampal tail at 6 months post-ECT associates with improved cognitive performance ($p_{FDR} < 0.05$, $N = 24$). Finally, patients performing worse on verbal fluency tests following treatment have greater electric field during ECT in the right hippocampal body ($p_{uncorrected} < 0.05$).
**Conclusions** The findings support that cognitive performance following ECT relates to macrostructural changes in the posterior cognitive hippocampus. Thus, there may be a threshold of ECT induced posterior hippocampal volumetric change, beyond which cognitive side effects occur.

## Plain language summary

Electroconvulsive Therapy (ECT) is a procedure that sends small electric currents through the brain and remains the most effective acute treatment for severe depressive episodes. However, we still do not fully understand how ECT works. Studies using brain scans (MRI) before and after ECT have shown that a part of the brain called the hippocampus often becomes larger after treatment. However, the clinical relevance of the volumetric change remains unknown. In this study, we looked at whether the increase in hippocampus size is linked to cognitive side effects. We found that a larger hippocampal volumetric increase after ECT was associated with reduced performance in verbal fluency tests, which measures our ability to rapidly produce words. These results suggest that big changes in the hippocampus after ECT may be related to short-term cognitive side effects.

Major depression is a leading cause of years lived with disability worldwide[1], with approximately 280 million of the world's population currently affected. Although several pharmacological and psychological treatments are available, electroconvulsive therapy (ECT) is still considered the most efficacious acute treatment option for moderate to severe depressive episodes[2,3]. Despite its robust clinical efficacy, its use is limited partly due to reports of cognitive side effects, including transient impairments of attention, executive functions, and memory[4]. Thus, unraveling the mechanisms of ECT-related cognitive side effects may lead to improved neurostimulation therapies that, in addition to being clinically efficient, ensure further improvements in cognitive safety.

One of the most consistently reported findings in ECT neuroimaging studies is transient hippocampal volumetric enlargement[5–9]. The finding is corroborated by recent meta-[8] and mega-[7] analyses reporting volumetric expansion up to 4–5% post-treatment, and suggestions of dose-response causative relationships. Moreover, electroconvulsive stimulation (ECS), a preclinical model of ECT, induces neuroplastic processes in the hippocampus[10–12], which may explain the transient volumetric increase reported in clinical studies. However, these neuroplastic processes likely affect the microstructural organization of this anatomical region, which may disturb hippocampal-dependent cognitive processes. Consistent with this notion, recent neuroimaging studies have demonstrated an association

✉ e-mail: olga.ousdal@uib.no

between global hippocampal volumetric expansion and procedure-related cognitive impairments[13–16]. Moreover, the strength of the hippocampal electric field (EF), which reflects the amplitude of the electric pulse as well as patient-specific anatomy, relates to both the hippocampal volumetric increase and changes in cognition[17].

Notably, not all studies have found an association between hippocampal volumetric change and cognitive side effects[17–19]. This may be due to clinical and methodological differences between studies, as well as limited sample sizes. In addition, most studies do not consider that the hippocampus is structurally and functionally organized along its longitudinal axis, with spatial gradients of afferent and efferent connections along the head, body, and tail divisions[20]. Accordingly, the precise location of ECT-mediated effects may determine side effects due to different downstream target regions[21]. Studies tracking the association between hippocampal long-axis changes and ECT-mediated cognitive side effects are, however, scarce.

The present study uses data from the Global ECT MRI Research Collaboration (GEMRIC) to address the association between hippocampal volumetric enlargement and cognitive changes following ECT. We focus on verbal fluency, since treatment-related changes in verbal fluency performance have been related to the strength of the hippocampal EF as well as hippocampal structural change[16,17]. With the recent advances in automatic hippocampal segmentation, it is now possible to obtain reliable volumetric estimates of hippocampal head, body, and tail[22]. Using these refined automatic segmentation protocols and the largest and most geographically diverse study sample to date, we investigate whether the association between ECT-related changes in verbal fluency and hippocampal volumetric increase was global or regionally specific. Based on recent findings suggesting that the anterior (i.e., head) region subserves affective and stress regulation functions, and the posterior (i.e., body and tail) regions are more involved in episodic memory and other cognitive processes[20,23], we hypothesize that volumetric changes confined to the posterior hippocampus would be associated with changes in verbal fluency performance following treatment. Finally, we test whether the strength of the estimated regional or global hippocampal EF is related to hippocampal volume change, and/or cognitive changes, thus providing a mechanistic explanation of the findings. We find that a greater volume increase in the posterior hippocampus is associated with reduced cognitive performance after ECT. However, the cognitive side effects improve as volumes normalize over time.

## Methods
### Participants
The present Global ECT-MRI Research Collaboration (GEMRIC) dataset (datarelease 3.2, DOI 10.17605/OSF.IO/YP2G4) included data from 22 study sites with neuroimaging and clinical data from 435 patients with uni- or bipolar depression as well as 119 healthy controls. Permission to use the GEMRIC dataset for the present work was given in accordance with the data sharing agreement for the GEMRIC study during the annual GEMRIC meeting in October 2020. More information regarding clinical and demographic characteristics of the participants is presented in Table 1 and Fig. S1. Participants received clinical, cognitive, and imaging assessments before (within one week before the first ECT session) and after the ECT index series (within 1–2 weeks after treatment completion), except for one site that scanned before and after the completion of nine ECT sessions. Healthy control participants were similarly scanned at two time points without receiving ECT treatment in between. In a subsample of patients, we also had long-term follow-up MRI data acquired 6 months following the completion of the ECT index series. Depression symptom severity was assessed with the Montgomery-Aasberg Depression Rating Scale (MADRS) or the Hamilton Depression Rating Scale converted to MADRS using a validated equation[24]. Cognitive data were generally collected before and after the ECT index series, except for a small subsample that also had a 6-month follow-up assessment. The cognitive assessment varied across sites and included tests of memory, attention, cognitive flexibility, and verbal fluency. Here, we examined two types of verbal fluency, specifically category and letter

verbal fluency (see Table S1), for which pre-post data were available for 124 of the patients and 24 patients at 6 months follow-up. For completeness of analyses, we also investigated treatment-related changes in episodic memory using the Hopkins verbal learning test, for which pre-post data were available for 42 patients.

Most patients used concurrent psychotropic medication, and a list of the medications is provided in Table S2. All participating sites obtained approval from their local ethics committee or institutional review board. All participants provided written informed consent after receiving study information. Each individual site has permission for data sharing of de-identified data to the GEMRIC study. The centralized mega-analysis was approved by the Regional Ethics Committee South-East Norway (#2018/769), and the study was conducted in accordance with the Declaration of Helsinki.

### Image acquisition and postprocessing
The image processing methods have been detailed previously[25]. In brief, structural T1-weighted MRI scans were acquired at each site, and the resulting DICOM images were transferred to a common data portal for analyses. The MRI images were acquired on 1.5 (2 sites) or 3 T (20 sites) scanners and had a minimum resolution of 1.3 mm in any direction (see Table S3). First, images were corrected for scanner-specific gradient non-linearity. Next, cortical parcellation and subcortical segmentation were performed using the longitudinal FreeSurfer recon-all stream (version 7.1, https://surfer.nmr.mgh.harvard.edu/).

In line with previous work from the GEMRIC, we adapted the quality control procedure from the Enhancing NeuroImaging Genetics through Mega-Analysis (ENIGMA) to identify potential outliers from the FreeSurfer version 7.1 longitudinal whole hippocampus segmentation (http://enigma.ini.usc.edu/protocols/imaging-protocols). A hippocampal volume was considered a statistical outlier if the volume estimate exceeded 2.7 standard deviations from the global mean. Identified outliers were inspected manually by a neuroimaging expert to determine segmentation errors. We reran all analyses without outliers caused by segmentation errors.

Hippocampal head, body and tail volume estimates were obtained by running the longitudinal hippocampal subfields segmentation algorithm as part of FreeSurfer v7.1[22,26]. This algorithm uses Bayesian statistics together with a hippocampal atlas obtained through manual delineation of ultra-high-resolution images of ex vivo hippocampal tissue[26].

Although the mode of electrode placement differed across sites, one electrode was always placed over the right hemisphere to ensure right hemispheric stimulation. Thus, we chose the right hippocampus, including its head, body, and tail subdivisions, for primary analyses addressing the association between volume change, individual estimated EF strength and cognitive side effects. Results for the left hippocampus are, however, presented in the Supplementary Results. We used the percent volume change relative to pre-ECT volume ($\Delta Vol_{r\text{-}hippocampus}/Pre\text{-}ECT\text{-}Vol_{r\text{-}hippocampus}$) as our within-subject assessment of longitudinal volume change.

### Electric field (EF) modeling
Realistic Volumetric-Approach to Simulate Transcranial Electric Stimulation (ROAST) v3.0[27] was used for estimation of the EF in the brain generated during the ECT treatment. The individual MRI image was segmented into five different tissue compartments (white matter, gray matter, cerebrospinal fluid, bone and scalp), and conductivity was assigned to each of the different compartments. From the segmented MRI, a three-dimensional tetrahedral mesh model of the head was built. Next, virtual electrodes of 5 cm diameter were placed with an automatic procedure over FT8 and C2 for right uni-lateral or FT8 and FT9 for bitemporal electrode placements from standard EEG locations offered in ROAST options. We calculated the EF potential using the finite element method to solve the Laplace equation with unit current on the electrodes, and this was subsequently scaled to the current amplitude of the two devices used (Thymatron 900 mA, Mecta Spectrum 800 mA). Then, the negative gradient of the EF potential generated a voxel-wise EF distribution map in each subject. The average EF across the whole

**Table 1 | Clinical and demographic characteristics of the sample**

| Participant characteristics | Mean | SD | N[a] |
|---|---|---|---|
| **Patients** | | | |
| Age, years | 54.0 | 15.5 | 434 |
| Sex, females (%) | 60.1 | | 435 |
| Baseline MADRS score | 25.5 | 6.9 | 418 |
| Post-treatment MADRS score | 9.7 | 7.8 | 415 |
| Duration of episode, months | 17.1 | 32.7 | 239 |
| No. of ECTs | 12.4 | 5.4 | 420 |
| No. of ECTs, RUL[b] | 12.1 | 5.1 | 291 |
| No. of ECTs, BL[b] | 14.6 | 6.1 | 129 |
| No. of ECTs remitters | 11.3 | 4.9 | 251 |
| No. of ECTs nonremitters | 14.2 | 5.8 | 164 |
| Baseline lvf score[c] | 20.7 | 17.3 | 127 |
| Post-treatment lvf score | 28.7 | 15.2 | 65 |
| Baseline cvf score[d] | 19.9 | 12.0 | 173 |
| Post-treatment cvf score | 23.1 | 12.3 | 107 |
| **Controls** | | | |
| Age, years | 47.0 | 14.6 | 119 |
| Sex, females (%) | 60.5 | | 119 |

*SD* standard deviation, *ECT* electroconvulsive therapy, *MADRS* Montgomery and Aasberg depression rating scale, *lvf* letter verbal fluency, *cvf* category verbal fluency.

[a]Due to missing data for some variables, the number of participants varies.
[b]Some participants received more than one mode of electrode placement.
[c]Note that some participants only completed the baseline or the post-treatment letter fluency tests.
[d]Note that some participants only completed the baseline or the post-treatment category fluency tests.

hippocampus and its three longitudinal subregions was calculated for each individual based on the FreeSurfer segmentations.

## Statistics and reproducibility

Individual subject-level data from 22 sites were available, and a series of General Linear Models (GLM) were conducted in R. Separate GLMs were performed for total hippocampus and the long-axis subregions. We tested for group differences (i.e., patients vs healthy controls) in percentage volume change and the association between hippocampal volume change and the estimated hippocampal EF. Next, we examined the associations between total or subregional hippocampal volume change and change in category or letter verbal fluency performance following treatment. Besides testing each hippocampal long-axis subregion separately, we also conducted two GLMs of change in category or letter verbal fluency against the volume change of all hippocampal long-axis subregions simultaneously. Next, we assessed the relationship between the estimated total or subregional hippocampal EF and pre-post changes in category or letter verbal fluency performance. Finally, to test whether the associations between gray matter volume change and cognitive outcome or the associations between estimated regional EF and cognitive outcome could be extended to other subcortical gray matter regions, we performed similar explorative analyses for the right amygdala. Age, sex, site and number of ECTs were included as covariates in all models, except for the group comparison that did not contain the number of ECTs. For the EF analyses, electrode placement (i.e., right unilateral, bilateral or mixed) was additionally included as a covariate, and the analyses of verbal fluency performance also controlled for the respective baseline verbal fluency scores.

Twenty-four of the patients had repeated the letter verbal fluency test at 6 months follow-up. Hence, GLMs were used to explore whether the long-term volumetric normalization (i.e., reduction) scaled with the improvement in letter fluency performance from post-index to long-term follow-up while controlling for age, sex, site, number of ECTs and the respective baseline verbal fluency scores.

To rule out confounding effects of potential extreme values on our results, we excluded participants with verbal fluency performance, estimated EF, or percentage hippocampal volume change values > |3| SD from the group mean from the statistical analyses. We calculated partial eta squared ($\eta_p^2$) as our effect size for dependent variables of interest for all GLMs, as it determines how large an effect the predictor variable had on the dependent variable. T-tests were two-tailed. The Benjamini–Hochberg false-discovery rate correction at $q = 0.05$ was used to correct for multiple comparisons.

## Reporting summary

Further information on research design is available in the Nature Portfolio Reporting Summary linked to this article.

## Results

### Clinical results

There was a significant decrease in MADRS scores following the ECT index series (MADRS baseline (mean ± SD): 25.5 ± 6.9, MADRS post-index: 9.7 ± 7.8, paired *t*-test: t = 31.30, $p < 0.001$). For the 124 patients completing a verbal fluency test before and after treatment, there were no significant changes in the category (t = 0.46, $p = 0.64$, N = 107) or the letter (t = 1.35, $p = 0.18$, N = 65) verbal fluency performance at the group level. Notably, 48 of the patients completed both tests. Importantly, pre-post changes in performance were unrelated for the two tests (r = 0.19, $p = 0.19$), thus reduced performance on the category verbal fluency was not related to a reduction in performance on the letter verbal fluency following treatment. Changes in clinical response were not associated with changes in category (r = −0.11, $p = 0.25$) or letter (r = −0.15, $p = 0.26$) verbal fluency.

### Right hippocampal volumetric change and regional electric field

In our primary analyses, we assessed group differences in right total and subregional hippocampal volumetric changes while controlling for age, sex and site. The analyses revealed significant volumetric enlargements of all right hippocampal subregions in the patients following the ECT index series (head: t = 9.05, $p_{fdr} < 0.001$, Cohen's d = 1.1; body: t = 6.35, $p_{fdr} < 0.001$, Cohen's d = 0.8; tail: t = 2.78, $p_{fdr} = 0.006$, Cohen's d = 0.4; total hippocampus: t = 9.07, $p_{fdr} < 0.001$, Cohen's d = 1.2). A within-group comparison of volumetric changes in patients confirmed pre-post ECT volumetric enlargements (Table S4). No significant changes were observed in healthy control participants (Table S5). Pairwise comparisons of the volumetric change in right hippocampal head, body, and tail revealed that the volumetric expansion of the right hippocampal head exceeded the volumetric enlargements of the right hippocampal body (t = 5.28, $p_{fdr} < 0.001$) and tail (t = 7.54, $p_{fdr} < 0.001$). Finally, the volumetric change of the right hippocampal body exceeded the volumetric change of the right hippocampal tail (t = 3.35, $p_{fdr} < 0.001$, Figs. 1a, b, S2).

Next, we tested the association between treatment-related volumetric changes and the strength of the anatomical corresponding estimated EF while controlling for age, sex, site, number of ECTs and electrode placement. In line with previous work from the GEMRIC[28], we observed no association between right hippocampal volumetric change and the estimated right hippocampal EF strength (t = −0.51, $p_{fdr} = 0.9$, $\eta_p^2 = 0.0007$). Similarly, there were no significant associations between hippocampal head, body or tail volumetric changes and the strength of the corresponding regional EFs (all $p > 0.05$). Interestingly, the estimated regional EF of the hippocampal body exceeded the estimated EF of the hippocampal head (t = 14.49, $p_{fdr} < 0.001$) and tail (t = 31.10, $p_{fdr} < 0.001$) (Fig. 1c). Please see Supplementary Results for analyses of left hippocampal volume change against left hippocampal EF.

### Right hippocampal volumetric change and cognitive side effects

We explored whether total or subregional hippocampal volume change was related to changes in verbal fluency performance while controlling for age, sex, site, number of ECTs and the respective baseline verbal fluency scores. The results revealed a significant negative association between change in hippocampal body volume following the index series and change in category

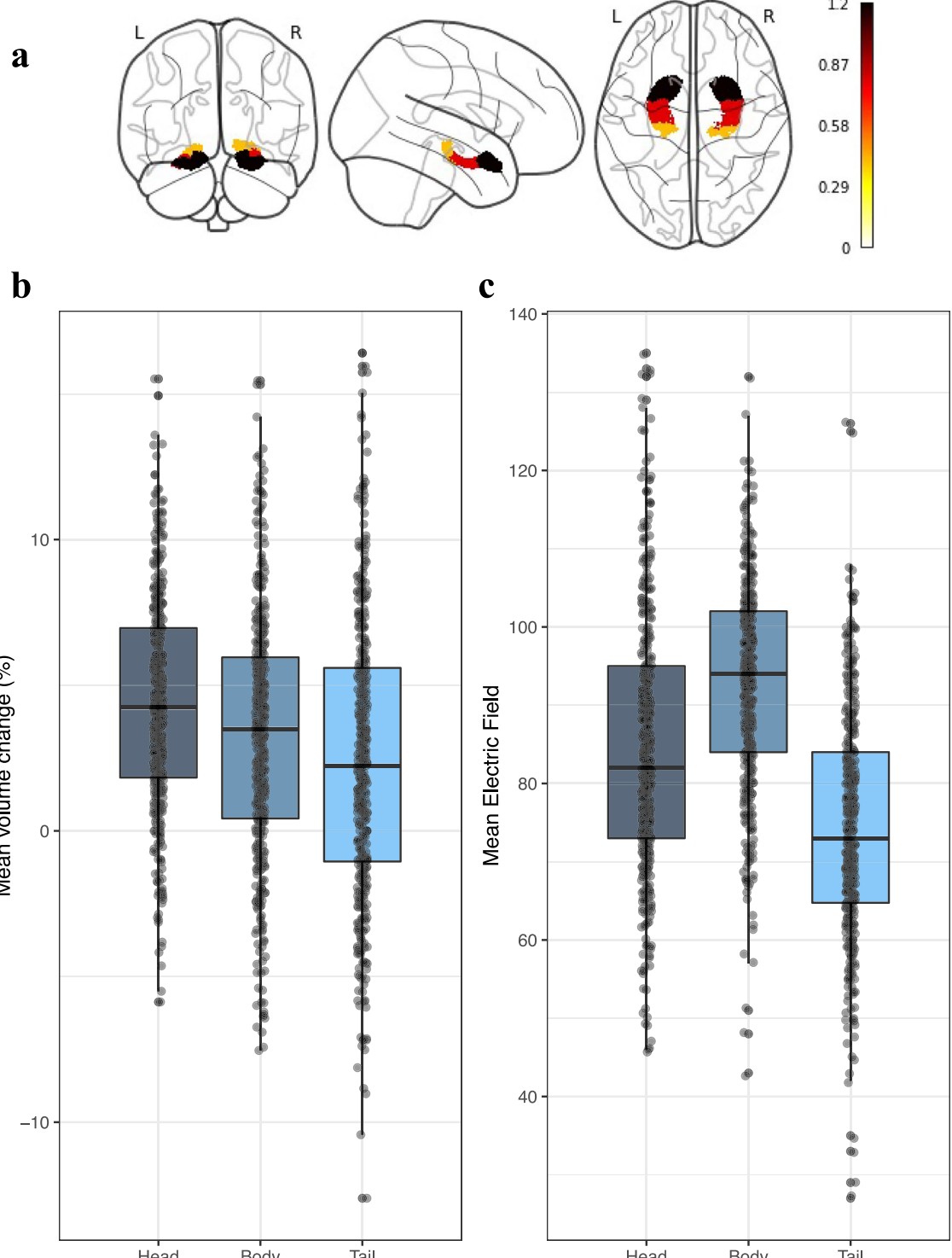

**Fig. 1 | Hippocampal longitudinal axis volumetric changes following electro-convulsive therapy and the hippocampal electric field strengths. a** Graphical illustration of right hippocampal head, body and tail percentage volumetric changes. The colors refer to Cohen's d effect sizes as coded in the bar to the right of the images. **b** Boxplot depicting percentage pre-post volumetric changes ($\Delta Vol_{hippocampus}$/Pre-

ECT-$Vol_{hippocampus}$) of right hippocampal head, body and tail ($N = 435$). **c** Boxplot depicting mean regional estimated Electric Field strength of right hippocampal head, body and tail ($N = 402$). Boxplots show the median and interquartile range (IQR). Whiskers extend to $1.5 \times$ IQR, with outliers plotted individually.

verbal fluency performance (t = −3.12, $p_{fdr}$ = 0.01, $\eta_p^2$ = 0.1, Fig. 2), suggesting that greater volumetric expansion was associated with a worsening in cognitive performance. The association remained significant after adjusting for total hippocampal volumetric change (t = −2.84, $p$ = 0.006, $\eta_p^2$ = 0.08). We ensured no multicollinearity of these models by inspecting

variance inflation factors, which all remained below 2. Using the same statistical framework, we also found a significant negative association between change in letter verbal fluency and the volumetric change of the hippocampal body (t = −2.66, $p_{fdr}$ = 0.04, $\eta_p^2$ = 0.1, Fig. 2) and tail (t = −2.32, $p_{fdr}$ < 0.05, $\eta_p^2$ = 0.09, Fig. 2). After adjusting for total

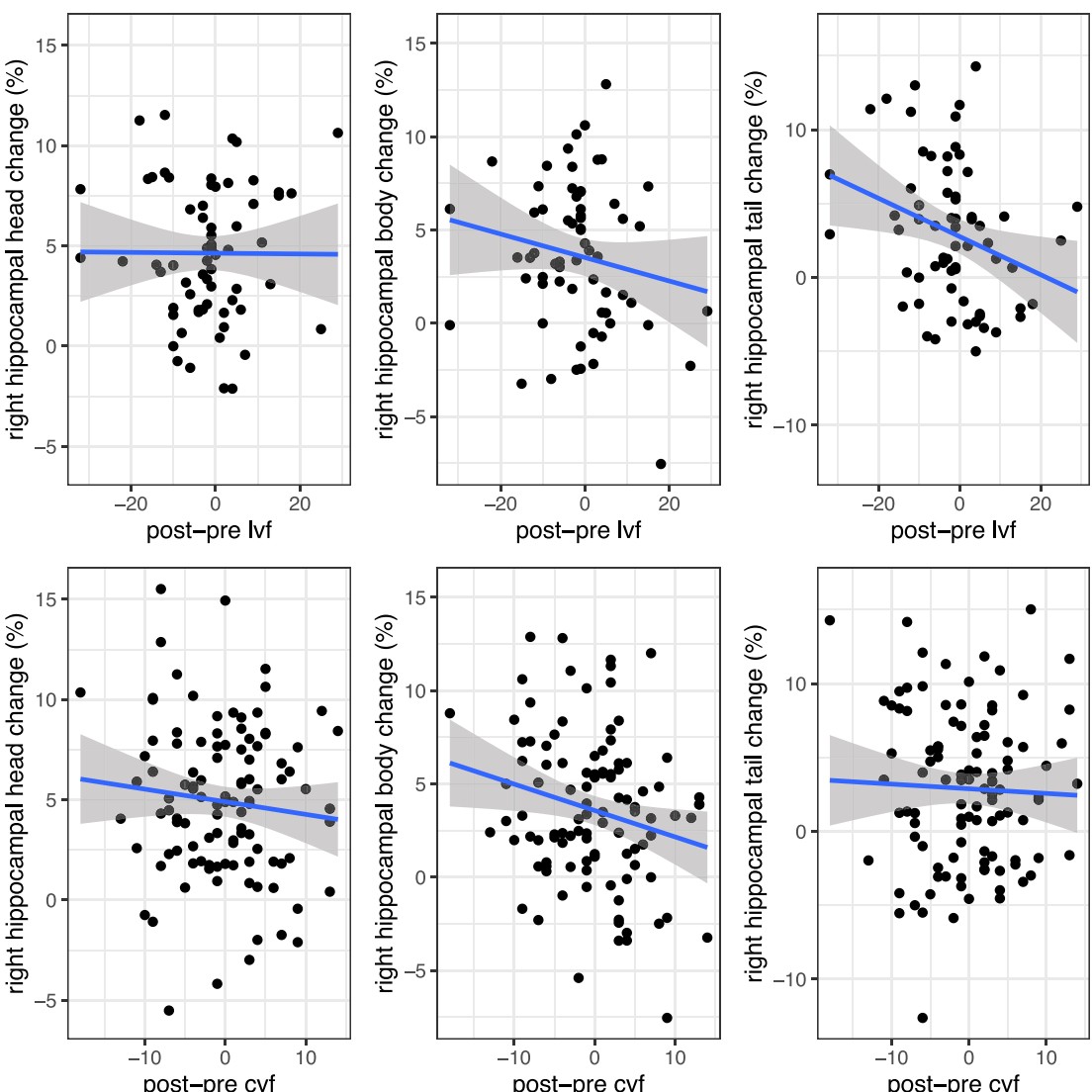

**Fig. 2 | The association between hippocampal longitudinal axis volumetric changes and cognitive performance following treatment completion.** Scatter plots of the association between percentage hippocampal head, body and tail volumetric changes following the index series ($\Delta$Vol$_{hippocampus}$/Pre-ECT-Vol$_{hippocampus}$) and the change (post – pre) in letter ($N = 65$) or category ($N = 107$) verbal fluency performance. The regression lines (with 95% confidence intervals shown as shaded areas) represent the relationship between the dependent and the independent variables calculated without covariates. lvf letter verbal fluency, cvf category verbal fluency, ECT electroconvulsive therapy.

hippocampal volumetric change, the association remained as a trend for the hippocampal body ($t = -1.91$, $p = 0.06$, $\eta_p^2 = 0.07$), but not for the hippocampal tail ($t = -1.20$, $p = 0.24$, $\eta_p^2 = 0.03$). The variance inflation factors remained below 2 for both models. Please see Supplementary Results for the association between left hippocampal volume change and verbal fluency performance, as well as the association between hippocampal volume change and episodic memory performance.

### Multiple linear regression models assessing changes in verbal fluency performance against the volume change of all hippocampal subregions

Besides testing each hippocampal long-axis region separately, we conducted two general linear models of change in category or letter verbal fluency against the volume change of all right hippocampal long-axis subdivisions simultaneously. Using backward elimination, the baseline model containing all regions of interest (ROIs) was successively compared to models with fewer and fewer ROIs to determine which long-axis regions best explained

the change in cognitive performance. Model comparisons utilized the Bayesian Information Criterion (BIC). For the change in category verbal fluency, the most parsimonious model included the hippocampal body only, and this model had a BIC = 686.61 (adjusted $R^2 = 0.26$, model $p$-value < 0.001), which was lower than the baseline model (BIC = 693.64, adjusted $R^2 = 0.26$, model $p$-value < 0.001). The difference in BIC score compared to the next-best model, which also included the hippocampal tail, was 2.5 (adjusted $R^2 = 0.27$, model $p$-value < 0.001). Similar results were obtained for letter verbal fluency, where the winning model, including the hippocampal body only, had a BIC score of 494.36 (adjusted $R^2 = 0.37$, model $p$-value < 0.001). This score differed by 2.4 BIC scores from the next-best (adjusted $R^2 = 0.37$, model $p$-value < 0.001) and 3.5 BIC scores from the baseline model (adjusted $R^2 = 0.39$, model $p$-value < 0.001).

### Hippocampal electric field and cognitive side effects

We analyzed the association between estimated total or subregional hippocampal EF and verbal fluency performance while controlling for age, sex,

site, number of ECTs, electrode placement and the respective baseline verbal fluency scores. There was no significant association between estimated total or subregional hippocampal EF and change in category or letter verbal fluency performance following treatment (all $p > 0.05$). However, if comparing patients performing worse on one or both tests with those experiencing no reductions in performance post-treatment, patients who performed worse ($N = 75$) had a trend toward greater EF in the hippocampal body (two-sample $t$-test: t = 2.16, $p = 0.03$, $p_{fdr} = 0.09$).

### Amygdala electric field, volumetric change and cognitive side effects

A paired $t$-test revealed significant volumetric enlargement of the right amygdala (t = 21.71, $p < 0.001$) from pre- to post-ECT. However, there were no significant associations between right amygdala volumetric change and the estimated right amygdala EF (t = −0.56, $p = 0.58$, $\eta_p^2 = 0.001$). Finally, there were no significant associations between right amygdala volume change or right amygdala EF and changes in category or letter verbal fluency performance (all $p > 0.05$).

### Long-term hippocampal volume reduction and improvement in cognitive performance

In a subsample of the patients ($N = 24$), we tested if the long-term normalization (i.e., reduction) of hippocampal volumetric change scaled with long-term improvement in letter verbal fluency performance while controlling for age, sex, site, number of ECTs and the baseline letter verbal fluency scores. Although the sample size was limited, there was a significant negative association between post-index to long-term volumetric change of the hippocampal tail (t = −2.8, $p_{fdr} < 0.05$, $\eta_p^2 = 0.3$, Fig. S3) and post-index to long-term performance on the letter verbal fluency test. Thus, participants experiencing the greatest volumetric reduction also improved the most in letter fluency performance from post-index to long-term follow-up.

## Discussion

In this study, we investigated the association between ECT-related cognitive side effects and hippocampal volumetric changes in the largest and most geographically diverse sample to date. Using measures of category and letter verbal fluency, we demonstrate an association between the individual change in verbal fluency performance and the volumetric enlargement of the right hippocampal body and/or the tail. Thus, greater volumetric expansion of the posterior hippocampus following treatment was associated with a worsening in verbal fluency performance. Furthermore, the degree of long-term reduction of right hippocampal tail volume was associated with improved letter verbal fluency performance from post-treatment to long-term follow-up. Finally, patients who experienced reduced verbal fluency performance following treatment may have had a higher EF in the right hippocampal body. Collectively, the findings suggest that excessive structural changes in the posterior hippocampus may be related to the cognitive side effects of ECT.

Considerable evidence supports that the hippocampus is not a uniform anatomical structure[20,23]. Indeed, preclinical research has shown that the anatomical connectivity and the gene expressions are topographically organized along an antero-posterior axis[20,23]. Thus, while the anterior hippocampus is predominantly connected with sensory cortical and limbic areas, including the amygdala, cortical regions like the retrosplenial and the anterior cingulate cortices show a posterior connectivity bias[20,29]. The connectivity patterns fit well with theoretical proposals suggesting that the anterior (i.e., head) regions subserve affective functions and regulate the stress response, while the posterior (i.e., body and tail) regions are heavily implicated in cognitive processes, including memory and spatial navigation[20,23]. In line with these theoretical proposals, a previous study suggested that structural changes confined to the hippocampal head are specifically related to the antidepressant response to ECT[21]. In contrast, changes in hippocampal body–angular gyrus functional connectivity assessed from resting-state functional MRI, and changes in hippocampal mean diffusivity assessed from diffusion weighted MRI were negatively

associated with verbal fluency performance following ECT[16,30]. We here extend these findings by showing that individual procedure-related changes in verbal fluency were associated with morphometric changes in the hippocampal body and tail, suggesting anatomic specificity of ECT-related cognitive side effects. The associations were discovered using a robust and conservative statistical framework, and the subregional specificity is visualized in Fig. 2. Of note, the regional specificity may explain why some previous studies found no association between hippocampal volumetric change and cognitive performance[18,19], as those studies investigated total hippocampal volume change. Indeed, verbal fluency relies on several cognitive functions, including memory-related processes, for which the posterior hippocampus plays a key role[20,23]. In addition, other brain regions important for verbal fluency, including frontal and temporal cortices, are predominantly interconnected with the posterior hippocampus. Thus, treatment-induced changes in posterior hippocampal structure may directly and indirectly, through an effect on downstream target regions, impact verbal fluency performance.

The neurobiological underpinning of the association between hippocampal volumetric change and cognitive side effects can be several. Preclinical animal studies have demonstrated that ECS stimulates neurogenesis in the dentate gyrus[10], which is further supported by changes in plasma neurogenesis markers in humans[31]. Beyond neurogenesis, several other neuroplastic processes and possibly also transient inflammatory responses could be related to both the volumetric increase and the cognitive side effects[32–35]. Thus, to further elucidate these mechanisms would require other neuroimaging modalities and preclinical work. Irrespective of the process, it is likely that this rapid modification of the hippocampus induced by repetitive ECT sessions may also transiently impact hippocampal-dependent cognitive functions. Indeed, the integration of newly formed neurons into the hippocampal trisynaptic circuitry temporarily impairs memory recall in animals[36], providing a direct link between hippocampal neuroplasticity and transient cognitive impairments. Of note, inflammatory and neuroplastic processes are likely to be important therapeutic ingredients of ECT; thus, the neurobiological underpinning of clinical response and side effects may possibly be shared[32]. If this is the case, then the procedure should be adjusted so that ECT induces sufficient stimulation to disrupt aberrant depressive circuitries but avoid excessive dosing that may also impact neural circuitries serving important cognitive functions[17,32].

Previous research has demonstrated that ECT parameters like electrode placement influence side effects[37]. More recent studies have also revealed that the ECT pulse amplitude, which determines the EF magnitude, impacts both the gray matter volumetric changes[17,28] and cognitive performance[17]. We found no association between right hippocampal volumetric change and the strength of the estimated hippocampal EF, including when investigating the total hippocampus or its long-axis subregions. This finding is in line with a previous study that used an overlapping GEMRIC patient sample[28]. Moreover, we found no linear association between change in verbal fluency performance and the total or the regional hippocampal EF. The lack of association between right hippocampal volumetric change and the estimated hippocampal EF may be due to a ceiling effect, where the EF surpasses a threshold at which there is no longer a dose-response relationship between the induced plasticity and volume change[28]. This ceiling effect may also prevent finding a dose-response relationship between regional EF and change in cognitive performance. Indeed, a previous study that randomized patients to different pulse amplitudes, and hence a larger distribution of EF strengths, did find a linear association between EF and verbal fluency performance, where a greater estimated EF significantly predicted cognitive worsening[17]. Thus, to further explore this association, we compared patients with and without worsened verbal fluency performance following treatment. Interestingly, worsening of performance was associated with greater estimated EF in the hippocampal body at an uncorrected significance threshold. Thus, patients developing cognitive side effects may be experiencing a higher hippocampal EF.

This study has some limitations. First, verbal fluency relies on a distributed fronto-temporal network, and hence it is not a direct test of hippocampal function[38]. However, the hippocampus plays a role in several of the cognitive processes implicated in verbal fluency, including episodic memory and executive functions, and there is an increasing number of studies relating verbal fluency performance to hippocampal structure and function[16,39]. Second, there was no verbal fluency data in the control group; thus, we were unable to control for test-retest effects. Third, our control group consisted of healthy participants, but future studies may benefit from also including depressed controls receiving alternative treatments to disentangle cognitive effects specific to ECT. Fourth, we were not able to control for electrical aspects (e.g., pulse width, pulse length and duration) of the ECT stimulation due to incomplete data. However, we acknowledge that measures like pulse width and stimulus intensity likely influence both the clinical response and cognitive side effects[37,40]. Moreover, we did not have information on continuation ECT, which could impact the verbal fluency performance at 6 months follow-up. Finally, we did not measure the effects of seizure characteristics, which have been shown to be a necessary component for clinical response[41], and may also be a contributor underlying volumetric change[42,43].

While hippocampal volumetric enlargement is consistently reported following ECT, the clinical relevance of the volumetric change has remained unclear. The findings of this study support that the volumetric changes of the hippocampal body and tail following ECT may be related to the procedure-associated cognitive side effects. Further, experiencing a decrease in verbal fluency performance following ECT was associated with a stronger regional EF in the hippocampal body at an uncorrected significance threshold. Thus, excessive posterior hippocampal structural change following ECT may not be beneficial and may indeed be directly related to the procedure-associated cognitive impairments. Future studies should address how ECT parameter characteristics and seizure collectively impact the clinical response and the side effects[44], preferably using prospective and harmonized study protocols.

## Data availability

The datasets generated and analyzed during the current study are available to members of the GEMRIC consortium, in accordance with the consortium's data sharing policies and subject to approval by the relevant ethics committee and the institution's data protection officer. Please contact associate professor Leif Oltedal regarding access to the data and request to join the collaboration. The source data for Figs. 1 and 2 are stored at a secure server (SAFE) at the University of Bergen, in accordance with ethical approvals. Access to the figure source data is granted to GEMRIC members upon request.

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

## Acknowledgements

This work was supported by the Western Norway Regional Health Authority (#911986 to K.J.O., # 912238 to L.O. and #912275 to O.T.O.), Fonds Wetenschappelijk Onderszoek (FWO-1168821N to M.L.), National Institute of Mental Health (MH119616 and MH120504 to M.A., MH128690 to R.E. and K.N., MH111826 to C.A.), the German Research Foundation (grant RE4458/1-1 to R.R.) and by the Federal Ministry of Education and Research (BMBF) of the German Center for Mental Health (DZPG; BMBF grant 01EE2305C to R.R.).

## Author contributions

O.T.O. conceptualized and coordinated the work in collaboration with L.O. O.T.O. analyzed the data and interpreted the results in collaboration with M.A., C.A. and L.O. M.L., A.A., F.B., J.A.C., M.C., N.C., U.D., A.D., L.E., R.E., K.H., R.H., M.J., M.K., T.K., K.L.N., P.N., N.O., R.R., D.R., A.S., D.S., P.S., C.S.M., A.T., F.t.D., I.D., M.U., L.v.D., P.v.E., G.v.W., J.v.W., M.V., J.V., B.S.C.W., Y.A., N.B., J.P., S.M., U.K., H.B., K.O., J.H. and Å.H. contributed data, as well as design of the project and critical revisions of the manuscript. All authors approved the final manuscript.

## Funding

## Competing interests

Joan A. Campordon is a member of the scientific advisory board of Hyka and Flow Neuroscience and has been a consultant for Mifu Technologies. Rene Hurlemann received lecture fees from Rovi and honoraria from Altheneum Consultation, Janssen und Rovi and Lundbeck. Antoine Yrondi received speaker's honoraria from Lundbeck, Janssen and Jazz and AstraZeneca. Shawn M. McClintock is a consultant to Pearson Assessment and receives royalties from Guilford Press, Inc. These organizations have no relation to the data or the results presented in this manuscript. All other authors declare no competing interests.

## Additional information

Olga Therese Ousdal [1,2] ✉, Miklos Argyelan [3,4], Maarten Laroy [5], Amit Anand [6,7], Filip Bouckaert[8], Joan A. Camprodon [9], Marta Cano [9], Narcis Cardoner [10,11,12], Udo Dannlowski [13], Annemiek Dols[14,15], Louise Emsell [5], Randall Espinoza [16], Kaat Hebbrecht[17,18], René Hurlemann [19], Martin Jorgensen [20], Maximilian Kiebs[19,21], Taishiro Kishimoto[22], Katherine L. Narr [16], Pia Nordanskog[23,24], Nils Opel[25,26], Ronny Redlich[26,27,28], Didi Rhebergen[29,30], Alexander Sartorius [31], Didier Schrijvers[32,33], Pascal Sienaert[18], Carles Soriano-Mas[12,34,35], Akihiro Takamiya [22], Freek ten Doesschate[36], Indira Tendolkar[37], Mikel Urretavizcaya[12,34,38], Linda van Diermen [32,39], Philip van Eijndhoven[37], Guido van Wingen [40,41], Jeroen van Waarde [36], Mathieu Vandenbulcke[5,8], Joey Verdijk[36], Benjamin S. C. Wade [9], Yrondi Antoine[42,43], Njål Brekke[2], Joan Prudic[44], Shawn McClintock[45], Ute Kessler[46,47], Hauke Bartsch [2,48], Ketil Odegaard[46,47], Jan Haavik [1,47], Åsa Hammar[49,50,51], Christopher Abbott [52] & Leif Oltedal [2,46]

[1]Department of Biomedicine, The Faculty of Medicine, University of Bergen, Bergen, Norway. [2]Mohn Medical Imaging and Visualization Center, Department of Radiology, Haukeland University Hospital, Bergen, Norway. [3]Feinstein Institutes for Medical Research, Institute of Behavioral Science, Manhasset, NY, USA. [4]The Zucker Hillside Hospital, Glen Oaks, NY, USA. [5]KU Leuven, Leuven Brain Institute, Department of Neurosciences, Neuropsychiatry, Leuven, Belgium. [6]Harvard Medical School, Boston, MA, USA. [7]Brigham and Women's Hospital, Neuroscience Center, Boston, MA, USA. [8]Geriatric Psychiatry, University Psychiatric Center KU Leuven, Leuven, Belgium. [9]Division of Neuropsychiatry and Neuromodulation, Department of Psychiatry, Massachusetts General Hospital, Harvard Medical School, Boston, MA, USA. [10]Sant Pau Mental Health Research Group, Institut d'Investigació Biomédica Sant Pau, Hospital de la Santa Creu I Sant Pau, Barcelona, Spain. [11]Department of Psychiatry and Forensic Medicine, Universitata Autònoma de Barcelona, Barcelona, Spain. [12]CIBERSAM, Carlos III Health Institute, Madrid, Spain. [13]Institute for Translational Psychiatry, University of Muenster, Muenster, Germany. [14]Amsterdam UMC location Vrije Universiteit Amsterdam, Psychiatry, Neuroscience, Amsterdam, The Netherlands. [15]Department of Psychiatry, UMC Utrecht Brain Center, University Medical Center Utrecht, Utrecht, The Netherlands. [16]Departments of Neurology and Psychiatry and Biobehavioral Sciences, University of California, Los Angeles, Los Angeles, CA, USA. [17]University Psychiatric Center KU Leuven, Department of Psychiatry, Leuven, Belgium. [18]Academic Center for ECT and Neuromodulation (AcCENT) University Psychiatric Center, KU Leuven, Kortenberg, Belgium. [19]Department of Psychiatry and Psychotherapy, School of Medicine & Health Sciences, Carl von Ossietzky University of Oldenburg, Oldenburg, Germany. [20]Psychiatric Center Copenhagen and Department of Clinical Medicine, University of Copenhagen, Copenhagen, Denmark. [21]Department of Psychiatry and Psychotherapy, University Hospital Bonn, Bonn, Germany. [22]Hills Joint Research Laboratory for Future Preventive Medicine and Wellness, Keio University School of Medicine, Tokyo, Japan. [23]Center for Social and Affective Neuroscience (CSAN), Department of Biomedical and Clinical Sciences, Linköping University, Linköping, Sweden. [24]Department of Psychiatry in Linköping, Linköping, Sweden. [25]Department of Psychiatry, University of Jena, Jena, Germany. [26]Institute of Translational Psychiatry, University of Muenster, Muenster, Germany. [27]Department of Psychology, University of Halle, Halle, Germany. [28]German Center for Mental Health (DZPG), Halle-Jena-Magdeburg, Magdeburg, Germany. [29]Amsterdam Public Health Research Institute, Department of Mental Health, Amsterdam UMC, Amsterdam, The Netherlands. [30]GGZ Central Innova, Department of Research, Amersfoort, The Netherlands. [31]Department of Psychiatry and Psychotherapy, Central Institute of Mental Health (CIMH), Medical Faculty Mannheim, University of Heidelberg, Heidelberg, Germany. [32]Collaborative Antwerp Psychiatric Research Institute, University of Antwerp, Antwerpen, Belgium. [33]University Psychiatric Hospital Duffel, Duffel, Belgium. [34]Department of Psychiatry, Bellvitge University Hospital, Bellvitge Biomedical Research Institute- IDIBELL, Barcelona, Spain. [35]Department of Social Psychology and Quantitative Psychology, Universitat de Barcelona, Barcelona, Spain. [36]Department of Psychiatry, Rijnstate Hospital Arnhem, Arnhem, The Netherlands. [37]Donders Institute for Brain, Cognition and Behavior, Department of Psychiatry, Radboud University Nijmegen, Nijmegen, The Netherlands. [38]Department of Clinical Sciences, Bellvitge Campus, Universitat de Barcelona, Barcelona, Spain. [39]Psychiatric Hospital Bethanië, Zoersel, Belgium. [40]Amsterdam UMC location University of Amsterdam, Department of Psychiatry, Amsterdam, The Netherlands. [41]Amsterdam Neuroscience, Amsterdam, The Netherlands. [42]Service de Psychiatrie et Psychologie Médicale, Centre Expert Dépression Résistante, Foundation Fondamental, CHU, Toulouse, France. [43]Toulouse Neuroimaging Center, Université de Toulouse, INSERM, Toulouse, France. [44]Department of Psychiatry, Colombia School of Medicine, New York, NY, USA. [45]Division of Psychology, Department of Psychiatry, UT Southwestern Medical Center, Dallas, TX, USA. [46]Department of Clinical Medicine, University of Bergen, Bergen, Norway. [47]Division of Psychiatry, Haukeland University Hospital, Bergen, Norway. [48]Department of Computer Science, University of Bergen, Bergen, Norway. [49]Department of Biological and Medical Psychology, University of Bergen, Bergen, Norway. [50]Department of Clinical Sciences Lund Psychiatry, Faculty of Medicine, Lund University, Lund, Sweden. [51]Department of Psychiatry, Skåne University Hospital, Lund, Sweden. [52]Department of Psychiatry, University of New Mexico, Albuquerque, NM, USA. ✉e-mail: olga.ousdal@uib.no

