## [Transparent Peer Review file · Communications Medicine]

Effects of Electroconvulsive Therapy on hippocampal longitudinal axis and its association with cognitive side effects

Corresponding Author: Dr Olga Ousdal

Version 0:

Reviewer comments:

Reviewer #1

(Remarks to the Author)

The hippocampal volumetric change is a well-documented phenomenon in ECT research. However, detailed studies are lacking. This well-written manuscript addressed that gap effectively. Here are my comments:

1. In both BL and RUL, the right electrode was placed over the right hemisphere to ensure stimulation. The authors focused on the right hippocampus, including its subdivisions. For RUL, since EF studies indicate significantly lower EF in the left hippocampus, I think it could serve as a natural within-subject control for examining EF effects on hippocampal volume changes. Could you conduct a subgroup analysis in the RUL group?
2. Given that 22 sites were enrolled, a mixed effect model could be an alternative statistical method?
3. Regarding the electric field (EF) results, since the EF was estimated and not directly measured in vivo, I recommend specifying this by using "estimated EF" for clarity.
4. In the discussion, the third paragraph, the authors mentioned that preclinical animal studies show ECS stimulates neurogenesis in the dentate gyrus. Beyond neurogenesis, other neuroplastic processes and possibly inflammatory responses could relate to volumetric increases and cognitive side effects. Some human ECT trials, such as those on neurogenesis (PMID: 38101468) and neuroinflammation (PMID: 37675893), also provide human in vivo evidence. Including these studies would enhance the discussion.

Reviewer #2

(Remarks to the Author)

This is a well-written study that focuses on specific macro-changes in the hippocampus during ECT for depression and associates this with one composite cognitive measure (verbal fluency). The study has gone one step further in unentangling specific changes to the hippocampal longitudinal macrostructures and finding clinically meaningful associations with cognitive deficits soon after index ECT and improvement at 6 months following the index course. Overall, this study will add further clarity to the role of specific changes to hippocampal structures and their associations with cognitive changes during ECT. It is likely to influence future work in optimising ECT stimulation paradigms to ultimately disconnect antidepressant effects of ECT from cognitive deficits.

Specific comments are made below. These are minor in nature.

Methods and Statistics

The multi-site nature of the study with a large sample of MRI data provides assurance for generalisability of the study. The data is complex and has many missing values. Nevertheless, the statistical methods are robust and thorough with GLM usage throughout. While reporting the values of P and its effect size are reported around partial eta square. CI around partial eta square is not computed. The authors are encouraged to comment on the relative appropriateness of measuring partial eta square as a measure of effect size.

Results

Page 11. It is better to be clear that it is the right hippocampus that is utilised for primary analysis and therefore for clarity change the sub-heading from "Hippocampal volumetric change and regional electric field" to right Hippocampal volumetric change and regional electric field. Specifying right hippocampus throughout the paragraph 2 of page 11 is also suggested to align it uniformly to the paragraph 3 where the EF of the right hippocampus is discussed. In Page 13, similar clarity in the heading with a change to "right hippocampal volumetric change and cognitive side effects" is suggested.

In Page 13, the main negative association between volumetric changes to the right hippocampal body and the verbal fluency tests have been demonstrated with a robust and conservative statistical framework. Moreover, the scatter plot (Figure 2) makes it visually clear regarding sub-regional specificity. These are to be commended.

Discussion.

In page 19, the authors have done well in highlighting pertinent limitations in this study. However, it is suggested that they also consider adding that they lack data on the electrical aspects of the ECT stimulation especially pulse width and stimulus intensity. Pulse widths, which carry packets of energy proportionate to the width of the pulses may influence differential cell body-axonal stimulations due to intrinsically smaller chronaxie values in axons compared to cell body (Ranck Jr, James B. "Which elements are excited in electrical stimulation of mammalian central nervous system: a review." Brain research 98.3 (1975): 417-440.).

The data is silent on continuation ECT after the index course. The authors are encouraged to consider this as a limitation in interpreting the verbal fluency data at 6 months (n=24).

Associate Professor Prashanth Mayur

Version 1:

Reviewer comments:

Reviewer #1

(Remarks to the Author)

I am very pleased with the authors' revision. Congratulations on the completion of this work.

Reviewer #2

(Remarks to the Author)

Thank you for your detailed responses. I have nothing further to comment.

Response to reviewers:

We are grateful to the reviewers for their thorough considerations of our manuscript and for their insightful considerations that have helped us improve our work. In the following sections, we respond to the reviewers' comments point-by-point and state revisions that have been made to the originally submitted manuscript. All updates can be viewed in the revised version of the manuscript where changes made from the originally submitted manuscript are marked with "track changes" in Word.

Reviewer #1 (Remarks to the Author):

The hippocampal volumetric change is a well-documented phenomenon in ECT research. However, detailed studies are lacking. This well-written manuscript addressed that gap effectively. Here are my comments:

1. In both BL and RUL, the right electrode was placed over the right hemisphere to ensure stimulation. The authors focused on the right hippocampus, including its subdivisions. For RUL, since EF studies indicate significantly lower EF in the left hippocampus, I think it could serve as a natural within-subject control for examining EF effects on hippocampal volume changes. Could you conduct a subgroup analysis in the RUL group?

Response: We thank the reviewer for raising this important question. In the Supplementary Results, we report a strong correlation between left hippocampal volume change and the strength of the corresponding left hippocampal EF in the total sample. As suggested by the reviewer, we have now also performed a subgroup analysis for the RUL group, and replicate previous findings of a correlation between left hippocampal volume change and left hippocampal EF also in this subgroup^{1, 2}. While the EF is likely to be significantly lower in the left hippocampus in RUL, it still surpasses the threshold necessary for inducing volumetric expansion^{1, 2}. In line with previous work from the GEMRIC¹, the associations remained after adjusting for site, number of ECTs and age. However, they did not remain after additionally controlling for sex (and electrode placement in the total sample). We have updated the *Volume changes, electric field and association with verbal fluency performance for the left hippocampus* section in the *Supplementary Results* accordingly:

Revision: *In line with previous work from the GEMRIC^{1, 2}, there was a strong correlation between left total hippocampal volume change and the strength of the corresponding left hippocampal electric field (EF) in the total sample ($r=0.27$, $t=5.55$, $p<0.001$) and in patients receiving right unilateral stimulation only ($r=0.25$, $t=4.50$, $p<0.001$). However, the association did not remain when adjusting for age, sex, site, number of ECTs and electrode placement (i.e. right unilateral, bilateral or mixed) (all $p> 0.05$). Moreover, there were no significant associations between hippocampal head, body or tail volumetric changes and the strength of the corresponding regional EFs (all $p> 0.05$) after adjusting for the covariates listed above. Total and/or regional left hippocampal EFs were not associated with changes in category or letter verbal fluency following treatment (all $p>0.05$).*

2. Given that 22 sites were enrolled, a mixed effect model could be an alternative statistical method?

Response: We agree with the reviewer that a linear mixed effect model could have been an alternative statistical method here. Our primary analyses tested the associations between pre-post ECT hippocampal volume change and changes in category or letter verbal fluency using general linear models adjusting for age, sex, site, number of ECTs and the respective baseline verbal fluency scores. We have now compared these general linear models with linear mixed effects models where site was added as a random effect, using ANOVAs. Despite added complexity, adjusting for site as a random effect did not lead to a significantly improved fit over the general linear model where site was a fixed effect (all p 's >0.05). Hence, we chose general linear models (the least complex and easier interpretable approach of the two) as our statistical framework throughout the manuscript.

3. Regarding the electric field (EF) results, since the EF was estimated and not directly measured in vivo, I recommend specifying this by using "estimated EF" for clarity.

Response: We thank the reviewer for this comment and have now revised the text accordingly.

4. In the discussion, the third paragraph, the authors mentioned that preclinical animal studies show ECS stimulates neurogenesis in the dentate gyrus. Beyond neurogenesis, other neuroplastic processes and possibly inflammatory responses could relate to volumetric increases and cognitive side effects. Some human ECT trials, such as those on neurogenesis (PMID: 38101468) and neuroinflammation (PMID: 37675893), also provide human in vivo evidence. Including these studies would enhance the discussion.

Response: We thank the reviewer for highlighting these very interesting studies. We have now revised paragraph three of the *Discussion* and cite these important studies.

Revision: *Preclinical animal studies have demonstrated that ECS stimulates neurogenesis in the dentate gyrus³, which is further supported by changes in plasma neurogenesis markers in humans⁴. Beyond neurogenesis, several other neuroplastic processes and possibly also transient inflammatory responses, could be related to both the volumetric increase and the cognitive side effects⁵⁻⁸.*

Reviewer #2 (Remarks to the Author):

This is a well-written study that focuses on specific macro-changes in the hippocampus during ECT for depression and associates this with one composite cognitive measure (verbal fluency). The study has gone one step further in unentangling specific changes to the hippocampal longitudinal macrostructures and finding clinically meaningful associations with

cognitive deficits soon after index ECT and improvement at 6 months following the index course.

Overall, this study will add further clarity to the role of specific changes to hippocampal structures and their associations with cognitive changes during ECT. It is likely to influence future work in optimising ECT stimulation paradigms to ultimately disconnect antidepressant effects of ECT from cognitive deficits.

Specific comments are made below. These are minor in nature.

Methods and Statistics

The multi-site nature of the study with a large sample of MRI data provides assurance for generalisability of the study.

The data is complex and has many missing values. Nevertheless, the statistical methods are robust and thorough with GLM usage throughout. While reporting the values of P and its effect size are reported around partial eta square. CI around partial eta square is not computed. The authors are encouraged to comment on the relative appropriateness of measuring partial eta square as a measure of effect size.

Response: We thank the reviewer for this comment. The partial eta square is telling us the proportion of total variation that is attributable to the predictor while excluding the other independent variables from the total variation. In other words, the partial eta square gave an estimate of how large an effect the predictor had on the dependent variable in our general linear models, and was therefore used as our effect size measure. We have now added a comment on the appropriateness of using partial eta square as our effect size in the *Statistical analyses* section.

Revision: *We calculated partial eta squared as our effect size for the predictors in the GLMs, as it determines how large an effect the predictor variable had on the dependent variable.*

Results

Page 11. It is better to be clear that it is the right hippocampus that is utilised for primary analysis and therefore for clarity change the sub-heading from “Hippocampal volumetric change and regional electric field” to right Hippocampal volumetric change and regional electric field. Specifying right hippocampus throughout the paragraph 2 of page 11 is also suggested to align it uniformly to the paragraph 3 where the EF of the right hippocampus is discussed. In Page 13, similar clarity in the heading with a change to “right hippocampal volumetric change and cognitive side effects” is suggested.

In Page 13, the main negative association between volumetric changes to the right hippocampal body and the verbal fluency tests have been demonstrated with a robust and conservative statistical framework. Moreover, the scatter plot (Figure 2) makes it visually clear regarding sub-regional specificity. These are to be commended.

Response: We agree with the reviewer about being more clear that it is the right hippocampus that was utilized for the primary analyses. We have now changed the suggested subheadings and text on page 10 and 11 to clarify this. We also highlight in the section two of

the *Discussion* that the findings were made through the use of a robust and conservative statistical framework and that the subregional specificity is supported by Figure 2.

Revision: *The associations were discovered using a robust and conservative statistical framework and the subregional specificity is visualized in Figure 2.*

In page 19, the authors have done well in highlighting pertinent limitations in this study. However, it is suggested that they also consider adding that they lack data on the electrical aspects of the ECT stimulation especially pulse width and stimulus intensity. Pulse widths, which carry packets of energy proportionate to the width of the pulses may influence differential cell body-axonal stimulations due to intrinsically smaller chronaxie values in axons compared to cell body (Ranck Jr, James B. "Which elements are excited in electrical stimulation of mammalian central nervous system: a review." *Brain research* 98.3 (1975): 417-440.).

The data is silent on continuation ECT after the index course. The authors are encouraged to consider this as a limitation in interpreting the verbal fluency data at 6 months (n=24).

Response: We thank the reviewer for these suggestions. We agree with the reviewer that the missing data on the electrical aspects of ECT and continuation ECT in the follow-up period are limitations to the study that should be addressed in the limitation section of the *Discussion*. We have made the following revision:

Revision: *Fourthly, we were not able to control for electrical aspects (e.g. pulse width, pulse length and duration) of the ECT stimulation due to incomplete data. However, we acknowledge that measures like pulse width and stimulus intensity likely influence both the clinical response and cognitive side effects^{9, 10}. Moreover, we did not have information on continuation ECT, which could impact the verbal fluency performance at 6 months follow-up.*

References

1. Argyelan M, Oltedal L, Deng ZD, Wade B, Bikson M, Joanlanne A et al. Electric field causes volumetric changes in the human brain. *Elife* 2019; 8: e49115.
2. Argyelan M, Deng ZD, Ousdal OT, Oltedal L, Angulo B, Baradits M et al. Electroconvulsive therapy-induced volumetric brain changes converge on a common causal circuit in depression. *Mol Psychiatry* 2024; 29(2): 229-237.
3. Madsen TM, Treschow A, Bengzon J, Bolwig TG, Lindvall O, Tingström A. Increased neurogenesis in a model of electroconvulsive therapy. *Biol Psychiatry* 2000; 47(12): 1043-1049.
4. Xie XH, Xu SX, Yao L, Chen MM, Zhang H, Wang C et al. Altered in vivo early neurogenesis traits in patients with depression: Evidence from neuron-derived extracellular vesicles and electroconvulsive therapy. *Brain Stimul* 2024; 17(1): 19-28.

5. Ousdal OT, Brancati GE, Kessler U, Erchinger V, Dale AM, Abbott C et al. The Neurobiological Effects of Electroconvulsive Therapy Studied Through Magnetic Resonance: What Have We Learned, and Where Do We Go? *Biol Psychiatry* 2021.
6. Yroni A, Sporer M, Pérán P, Schmitt L, Arbus C, Sauvaget A. Electroconvulsive therapy, depression, the immune system and inflammation: A systematic review. *Brain Stimul* 2018; 11(1): 29-51.
7. Cano M, Camprodón JA. Understanding the Mechanisms of Action of Electroconvulsive Therapy: Revisiting Neuroinflammatory and Neuroplasticity Hypotheses. *JAMA Psychiatry* 2023.
8. Xu SX, Xie XH, Yao L, Wang W, Zhang H, Chen MM et al. Human in vivo evidence of reduced astrocyte activation and neuroinflammation in patients with treatment-resistant depression following electroconvulsive therapy. *Psychiatry Clin Neurosci* 2023; 77(12): 653-664.
9. Loo CK, Schweitzer I, Pratt C. Recent advances in optimizing electroconvulsive therapy. *Aust N Z J Psychiatry* 2006; 40(8): 632-638.
10. Sackeim HA, Prudic J, Devanand DP, Kiersky JE, Fitzsimons L, Moody BJ et al. Effects of stimulus intensity and electrode placement on the efficacy and cognitive effects of electroconvulsive therapy. *N Engl J Med* 1993; 328(12): 839-846.